# A Bibliometric Analysis of Peer Assessment in Online Language Courses

**Yupeng Lin** and **Zhonggen Yu** *

Faculty of Foreign Studies, Beijing Language and Culture University, Beijing 100083, China
* Correspondence: yuzhonggen@blcu.edu.cn

**Abstract:** As a popular strategy in collaborative learning, peer assessment has attracted keen interest in academic studies on online language learning contexts. The growing body of studies and findings necessitates the analysis of current publication trends and citation networks, given that studies in technology-enhanced language learning are increasingly active. Through a bibliometric analysis involving visualization and citation network analyses, this study finds that peer assessment in online language courses has received much attention since the COVID-19 outbreak. It remains a popular research topic with a preference for studies on online writing courses, and demonstrates international and interdisciplinary research trends. Recent studies have led peer assessment in online language courses to more specific research topics, such as critical factors for improving students' engagement and feedback quality, unique advantages in providing online peer assessment, and designs to enhance peer assessment quality. This study also provides critical aspects about how to effectively integrate educational technologies into peer assessment in online language courses. The findings in this study will encourage future studies on peer assessment in online learning, language teaching methods, and the application of educational technologies.

**Keywords:** peer assessment; online language learning; educational technologies; bibliometric analysis; citation network analysis; visualization analysis

## 1. Introduction

Collaborative learning can enhance learning outcomes, cognitive abilities, and social skills (Laal and Ghodsi 2012). A growing size of evidence can be found in previous empirical studies and reviews to support the advantages of collaborative learning, or more specifically, peer assessment (Jung et al. 2021). Online learning has demonstrated its significant role and advantages, especially during the COVID-19 pandemic (Adedoyin and Soykan 2020). Online learning and peer assessment in various subjects continue to develop, inspiring integrative research to investigate peer assessment in online language courses and provide enhanced and diversified forms of it. With the development of information and technology, educational technologies have become popular, and more technologies have been introduced to teaching practice, as listed by Haleem et al. (2022). Although collaborative learning may take forms other than peer assessment, many recent studies focus on it in online education contexts, where educational technologies are usually integrated. Consequently, peer assessment in online language courses sparks constant academic interest and is now a critical component of studies on online education.

The significance of peer assessment in online language education is evident in its vast advantages when applied to distance education. For instance, peer assessment effectively promotes learners' writing skills, reflective thinking abilities, and problem-solving efficiency (Lin 2019; Liang and Tsai 2010). Implementing peer assessment activities in online language teaching also brings improved academic achievements compared with traditional feedback and offline courses, such as transferable skills, a better understanding of the assessment criteria, timely feedback, and constant learning and development (Adachi et al. 2018).

However, there are also problems with peer assessment in online language education. The quality of peer assessment may be limited due to students' expertise; how to enhance the students' engagement and self-efficacy in peer assessment is still not completely clear (Lin 2019; Adachi et al. 2018). There is still a controversy about peer assessment in online language courses, which requires researchers to continue further investigations. Although previous studies have provided reviews and comments on the existing studies and findings about this topic, little is known about the research trends and the whole citation network of the related literature. Therefore, a bibliometric and systematic citation network analysis is significant in pushing the frontiers of this topic.

In order to bridge the research gap and inspire future investigations on this topic, this study intends to conduct a bibliometric analysis of peer assessment in online language courses. In this article, we will first review literature related to this topic and then provide methods of a bibliometric analysis with VOSviewer and CitNetExplorer. Based on the literature search, we conducted visualization and citation network analyses where literature clustering generates further discussion. This bibliometric study focuses on the theoretical foundations, previous empirical studies, and the recent developments related to this topic. By investigating the existing literature, we aim to help understand how peer assessment in online language courses has been established and integrated into teaching practice. The findings of this study will provide a clearer overall view of this topic and help identify how the current findings may be further developed in future studies.

## 2. Literature Review

### 2.1. Introducing Peer Assessment to Online Language Courses with Interdisciplinary Research

Rising interest in peer assessment and online language courses has been revealed by researchers in recent years (Li et al. 2020). A considerable number of studies have combined these two aspects, investigating the implementation of peer assessment in online language courses (Lin 2019). The overall findings of most existing studies suggested positive outcomes: peer assessment in online contexts could enhance learners' language learning achievements (Liu et al. 2018; Ghahari and Farokhnia 2018). With the popularity of e-learning contexts, peer assessment has been actively applied. To illustrate this point, empirical results on this topic have been significantly enriched by technological advancements and applications since the advent of the 2020s. Examples can be found in further applications of video annotation tools, online teaching platforms to satisfy the educational needs in the post-COVID-19 era, and automated evaluation tools corresponding to e-learning environments (Fang et al. 2022; Shek et al. 2021). These studies contributed to understanding peer assessment in the era of online learning and teaching triggered by COVID-19 (Adedoyin and Soykan 2020).

Researchers in linguistics extended their interest to different language skills and learning contents in language courses, and they actively sought interdisciplinary methods and perspectives outside language education studies. Empirical evidence supporting peer assessment in online language education was provided from the existing literature, even if research interest in different language skills could vary. Among them, commonly investigated language skills included listening (Tran and Ma 2021), speaking (Nicolini and Cole 2019), and writing (Sun and Zhang 2022; Zhang et al. 2022). More integrative language courses included English for specific purposes (Salem and Shabbir 2022), academic writing (Topping et al. 2000; Cheong et al. 2022), and communication skills (Shek et al. 2021). The development of applied linguistics by involving theories and concepts from various subjects and research areas made interdisciplinary approaches increasingly crucial. The interactions between linguistic studies and theories in other research areas have conceived popular research methods and open-minded perspectives in current studies. Dominantly reflecting the interdisciplinary trends, psychological constructs were examined when educational technologies were introduced to language courses, shedding light on the effects of e-learning methods on students' motivation, engagement, self-efficacy, and cognitive load (Akbari et al. 2016; Lai et al. 2019; Yu et al. 2022a, 2022b).

*2.2. Theoretical Foundations of Peer Assessment and Reviews in Online Learning Contexts*

The idea of peer assessment was derived from early discussions on collaborative learning, and the subsequent studies continued extending the theories and the applications of peer assessment. Inspired by Piaget (1929), who suggested that collaborative learning and cognitive construction were related and developed together (Roberts 2004), Vygotsky started with his theories, indicating how learners would more easily learn knowledge and develop particular skills (Vygotsky and Cole 1978). However, the concept of peer assessment was not termed first until Keith J. Topping. Although Topping (2009) suggested earlier origins of peer assessment in educational contexts, researchers now dominantly attributed the blossoming studies related to this topic to Topping's systematic foundations of peer assessment (Lin 2019). In his earlier reviews of peer assessment, Topping predicted the rising trend of computer-assisted peer assessment (Topping 1998). In a series of articles and books, he reviewed and summarized the developments of peer assessment (Topping 1998, 2009, 2018), inspiring emerging research topics related to peer assessment in various educational contexts. Relying on the theoretical foundations, studies in recent years have contributed to new theories with growing empirical evidence in diversifying educational contexts.

Apart from the reviews of peer assessment from an educational perspective generally, existing reviews and meta-analyses of peer assessment in online language courses concentrated on relatively narrow scopes. For example, reviews and meta-analyses concentrated on particular language skills like argumentative writing (Awada and Diab 2021) or technology-assisted writing without involving peer assessment (Williams and Beam 2019). One problem was that these reviews and meta-analyses focused on particular aspects while leaving reviews from a general perspective a research gap. The existing reviews and meta-analyses needed a citation network analysis to include studies with peer assessment in online language education. The perspective of this review is an intermediate scope, not limited to language education in traditional classrooms but revealing the characteristics specific to peer assessment in online language education.

*2.3. Educational Technology Applications for Peer Assessment in Online Langauge Courses*

Although collaborative learning was proposed much earlier, introducing educational technologies to peer assessment benefited from rapid and revolutionary changes in the era of information and technology (Roberts 2005). Not specifically for peer assessment, educational technologies were introduced into language education for various purposes. As summarized by Haleem et al. (2022), in technology-enhanced language education studies, dominant educational technologies included at least the following: (1) quick assessment technologies, for example, automated writing evaluation tools (Nunes et al. 2022); (2) resources for distance learning, especially video-based instructions, such as Massive Open Online Courses (MOOCs) (Fang et al. 2022) and video conferencing (Hampel and Stickler 2012); (3) electronic books and digital reading technologies (Reiber-Kuijpers et al. 2021); (4) broad access to the most up-to-date knowledge and enhanced learning opportunities; (5) mobile-assisted language learning (as reviewed by Burston and Giannakou 2022); and (6) social-media-assisted language learning (Akbari et al. 2016). With diversified techniques and designs, technology-enhanced language learning in recent years was established as an emerging, rapidly developing, interdisciplinary, and critical research area.

With the emergence of educational technologies, the frontiers in studies on peer assessment in online language education witnessed the welcoming integration of education technologies. Many empirical studies in more recent years closely turned to popular educational technologies, and were eager to discover the application of peer assessment in online language courses. Video-based peer feedback was considered an efficient tool to provide emotionally supportive feedback for language learners with an enhanced sense of realistic perception using virtual reality (Chien et al. 2020). Compared with traditional forms of written feedback, video peer feedback improved learners' Chinese-to-English translation performance (Ge 2022). Digital note-taking technologies demonstrated unique

advantages in addressing learners' special needs when suffering from language-related disabilities (Belson et al. 2013). However, the educational technologies applied to peer assessment in online language courses still comprised a small proportion of dominant educational technologies assisting language learning.

### 2.4. Research Questions

Based the research gaps revealed in the previous sections, we would like to answer the following research questions (RQs):

RQ1: In terms of the number of publications, research areas, and distribution of publication journals, what are the publication trends of studies on peer assessment in online language courses?

RQ2: What are the top authors, keywords, countries, and organizations in the studies on peer assessment in online language courses?

RQ3: What are the most studied language skills in the existing literature related to peer assessment in online language courses?

RQ4: What are the commonly investigated interdisciplinary topics related to peer assessment in online language courses?

RQ5: How are the theoretical foundations of peer assessment commonly cited in current studies on the application to online language courses?

RQ6: How can educational technologies be integrated into peer assessment in online language courses?

First, understanding the publication trends and the most cited study contributors would be a basic and essential way to follow the development and approach the frontiers of this topic. Based on the existing literature, this study would provide the primary bibliometric results of studies on peer assessment in online language courses to answer RQ1 and 2. To provide more details of the existing literature, RQ3 would consider studies within language education research and identify the dominant research topics. In contrast, RQ4 would take perspectives outside language education, investigating the interdisciplinary approaches to linguistic and educational research. We would also explore how the theoretical foundations were cited in current studies and examine whether new sources of updated theoretical foundations in the existing literature have arisen. Therefore, RQ5 was proposed and would be answered based on our citation network analysis. In properly enhancing the effectiveness of language teaching and learning in technology-assisted environments, there are opportunities, but also challenges. By exploring peer assessment in online language courses, we would like to provide some suggestions about how to integrate educational technologies into peer assessment in online language courses by answering RQ6.

## 3. Methods

### 3.1. Literature Search and Result Analysis on Web of Science

We searched the related literature on Web of Science (WOS), a worldwide literature search engine that provides access to databases and journals. More specifically, we selected the Core Collection of WOS, i.e., a selected collection of journals. The Core Collection comprises multiple indexes of high-quality journals, including Science Citation Index Expanded (2013 to present), Social Sciences Citation Index (2006 to present), Arts and Humanities Citation Index (2008 to present), and Emerging Sources Citations Index (2017 to present). We did not use Current Chemical Reactions (1985 to present) and Index Chemicus (1993 to present) due to their unrelated research areas to this study. Keywords were searched as follows: Peer assessment OR peer evaluation OR peer feedback OR peer review (Topic) AND online (Topic) AND writing OR speaking OR oral OR spoken OR listening OR reading OR language (Topic). The search results were filtered by selecting related research areas to concentrate the results on the studies of online language education, including "Education and Education Research", "Language and Linguistics", "Communication", "Knowledge Engineering and Representation", and "Translational Studies". WOS was used to analyze the search results. The analysis function integrated into the website would allow

a preliminary bibliometric analysis by counting results published in different years, WOS categories, and sources (journals). These three items would provide the distribution of year-based publications, the top 10 published categories, and the top 10 publication journals.

### 3.2. Visualization and Citation Network Analysis

We exported full records and cited references of the search results in plain text files for visualization analysis. This study used two popular computer programs for visualization and citation network analyses, i.e., VOSviewer (Van Eck and Waltman 2014) and CitNet-Explorer (Van Eck and Waltman 2010). VOSviewer allowed the visualization of authors, keywords, countries, and organizations based on the relationships between items according to the literature records. We generated the visualized maps by setting the minimum occurrence = 2 for each item and the lists that counted the citations and occurrences of authors, keywords, countries, and organizations. In order to offer representative results while covering a relatively large picture of studies on this topic, the top 20 authors, keywords, countries, and organizations with the highest citations or occurrences would be presented and analyzed in this study. Excluding online-first papers was required to avoid technical problems because publication years were not provided for the online-first papers to be analyzed by CitNetExplorer. Then, the rest of search results were used for a citation network analysis. We chose to exhibit the non-matching cited references in the software so that the entire network could be included. The clustering function in this software would categorize the results to show the citation network of the search results (Van Eck and Waltman 2017). The function of longest path analysis allowed examining how previous theories and findings were cited and developed by more recent studies via multiple times of citations. The longest paths would help identify the pioneering research where valuable foundations were set and inherited to current studies.

## 4. Results

### 4.1. Literature Search

On 16 November 2022, we searched the Core Collection of WOS. The filtered results by research areas included 484 records. To reveal publication trends (RQ1), we found the answers from figures and lists generated by the "Analyze Results" function on WOS. Figure 1 demonstrates the number of publications on peer assessment in online language courses between 2008 and 2022. Since 2008, the studies related to peer assessment in online language courses have been steadily increasing, especially with a rapid increase in 2019 and 2021. During the past 15 years, the current climax was in 2021, with 83 results. The year 2022, despite the incomplete counting, recorded N = 66. For all 484 results related to language and linguistic studies, the primary Web of Science categories were demonstrated in Table 1. The primary publication journals were shown in Table 2, where the top 10 journals have published 26.24% results (N = 127). Among the 484 results, 27 belonged to the online-first papers, which were excluded for citation network analysis using CitNetExplorer. The exported full records of the results after excluding online-first papers included 457 studies.

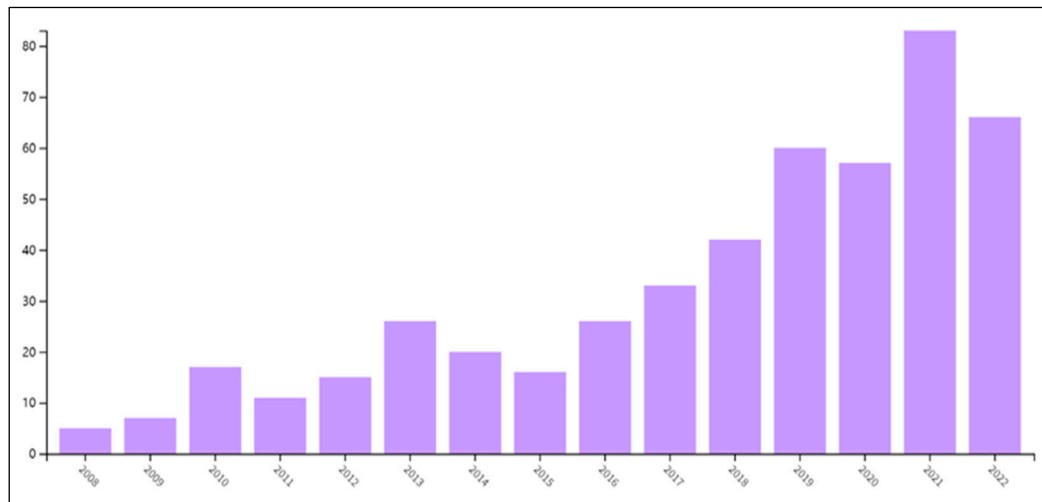

**Figure 1.** Publication trend of the studies on peer assessment in online language courses.

**Table 1.** The top 10 Web of Science Categories with most search results.

| Web of Science Categories | Results | Percentage |
|---|---|---|
| Education Educational Research | 312 | 64.46% |
| Linguistics | 83 | 17.15% |
| Language Linguistics | 49 | 10.12% |
| Computer Science Interdisciplinary Applications | 33 | 6.82% |
| Education Scientific Disciplines | 31 | 6.41% |
| Information Science Library Science | 16 | 3.31% |
| Psychology Multidisciplinary | 16 | 3.31% |
| Communication | 13 | 2.69% |
| Computer Science Information Systems | 12 | 2.48% |
| Psychology Educational | 9 | 1.86% |

**Table 2.** The top 10 journals in the search results.

| Web of Science Categories | Results | Percentage |
|---|---|---|
| Computers Education | 22 | 4.55% |
| Computer Assisted Language Learning | 19 | 3.93% |
| Interactive Learning Environments | 15 | 3.10% |
| Assessment Evaluation in Higher Education | 13 | 2.69% |
| Recall | 11 | 2.27% |
| System | 11 | 2.27% |
| Australasian Journal of Educational Technology | 10 | 2.07% |
| Educational Technology Society | 9 | 1.86% |
| Frontiers in Psychology | 9 | 1.86% |
| Asia Pacific Education Researcher | 8 | 1.65% |

*4.2. Visualization Analysis and Most-Cited Items*

In order to answer RQ2, the filtered studies with online-first papers were processed with VOSviewer, where 468 keywords were visualized in Figure 2. Keyword items were grouped by their connections into 17 clusters. Clusters were distinguished with different colors, and the connections between nodes demonstrated how the keyword items were combined in published articles. Table 3 displayed seven clusters that contained more than 30 keyword items, which amounted to 270 keywords (57.69% of all). The other 10 clusters were relatively small and less frequently used, so they might not reflect the academic interest related to this topic. Through the visualization analysis of the primary clusters, the representative keyword items in Table 3 and the salient ones in Figure 2

revealed the interest in specific research topics related to peer assessment in online language courses. The integration of educational technology into online language education was particularly demonstrated by various keyword items related to technologies. The keywords with the highest occurrences slightly varied in referring to "peer assessment". Although peer assessment belongs to collaborative learning strategies and teaching approaches, researchers used peer "assessment", "feedback", "evaluation", and "review". Regarding educational levels, "higher-education" was the 17th most used keyword. Popular keyword items also included different learning content, including "knowledge" and "skills", as well as psychological constructs, such as "motivation" and "perceptions".

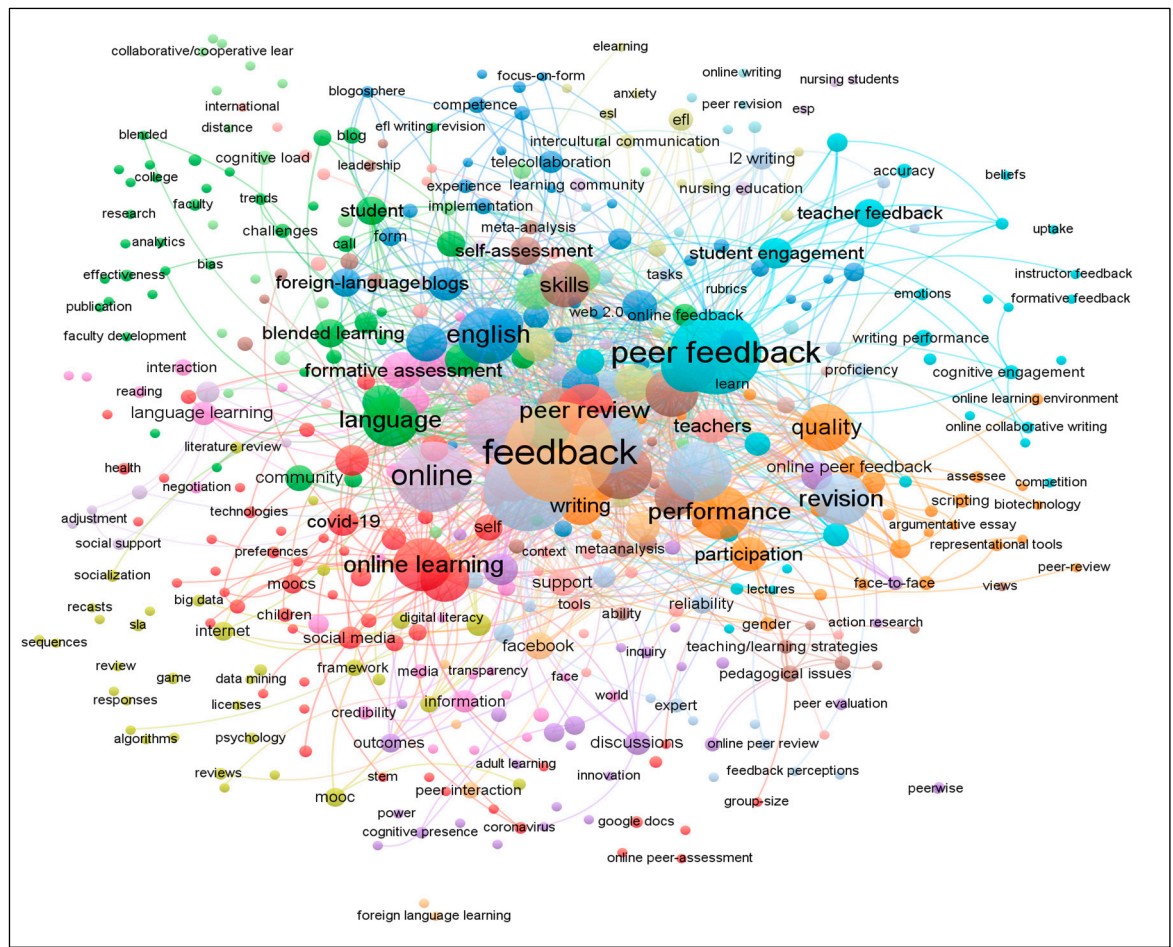

**Figure 2.** Visualization of the co-occurrence of keywords in the studies.

**Table 3.** Keyword item clustering in VOSviewer.

| Cluster No. | Color | Items (Percentage) | Representative Keyword Items with High Occurrences |
|---|---|---|---|
| 1 | 🟥 | 56 (11.97%) | efficacy, engagement, online learning, peer review, and strategies |
| 2 | 🟩 | 46 (9.83%) | language, academic writing, blended learning, and science |
| 3 | 🟦 | 39 (8.33%) | assessment, blog, communication, and English |
| 4 | 🟨 | 33 (7.05%) | internet, model, MOOC, framework, learning analytics, and psychology |
| 5 | 🟪 | 33 (7.05%) | computer-mediated communication, 2nd-language, EFL writing, |
| 6 | 🟦 | 32 (6.84%) | higher education, peer feedback, technology, and student perception |
| 7 | 🟧 | 31 (6.62%) | performance, participation, writing, and online peer feedback |

Similarly, we generated lists of 64 authors, 127 organizations, and 44 countries that occurred at least twice. In Tables 4 and 5, we listed the top 20 authors, organizations, and countries according to their citations, and listed the top 20 keyword items according to

their occurrences in the search results. Most studies were conducted in English-speaking countries, such as the US, the UK, and Australia. Many studies on peer assessment in online language courses were conducted in non-English-speaking countries. Countries such as China, Spain, and the Netherlands do not use English as their official language, but they have contributed to hundreds of publications on this topic.

**Table 4.** The top 20 authors and keyword items in the search results.

| Authors | Citations | Link | Keyword | Occurrences | Link |
|---|---|---|---|---|---|
| Schunn, Christian D. | 375 | 1 | feedback | 96 | 620 |
| Shih, Ru-Chu | 200 | 0 | peer feedback | 69 | 449 |
| Noroozi, Omid | 195 | 21 | students | 68 | 440 |
| Yang, Yu-Fen | 194 | 1 | online | 53 | 316 |
| Lee, Lina | 153 | 0 | education | 44 | 302 |
| Espasa, Anna | 108 | 3 | perceptions | 41 | 273 |
| Guasch, Teresa | 108 | 3 | PA | 40 | 241 |
| Mulder, Martin | 106 | 6 | impact | 37 | 289 |
| Liang, Jyh-Chong | 105 | 2 | English | 37 | 245 |
| Tsai, Chin-Chung | 105 | 2 | performance | 34 | 260 |
| Wang, Yanqing | 97 | 1 | online learning | 33 | 182 |
| Hatami, Javad | 83 | 7 | peer review | 33 | 170 |
| Bradley, Linda | 79 | 0 | revision | 32 | 232 |
| Biemans, Harm J. A. | 75 | 14 | language | 32 | 195 |
| Liu, Gi-Zen | 75 | 4 | technology | 31 | 224 |
| Yeh, Hui-Chin | 69 | 1 | knowledge | 31 | 213 |
| Barrett, Neil E. | 64 | 3 | higher-education | 28 | 204 |
| Mostert, Markus | 63 | 2 | quality | 27 | 201 |
| Snowball, Jen D | 63 | 2 | skills | 26 | 193 |
| Latifi, Saeed | 54 | 8 | motivation | 24 | 202 |

**Table 5.** The top 20 organizations and countries in the search results.

| Organizations | Documents | Citations | Strength | Countries | Documents |
|---|---|---|---|---|---|
| University of Canterbury | 2 | 418 | 0 | USA | 137 |
| University of Pittsburgh | 6 | 406 | 5 | China | 117 |
| University of Surrey | 2 | 244 | 1 | England | 31 |
| National Pingtung University Science and Technology | 2 | 200 | 0 | Spain | 30 |
| Tarbiat Modares University | 8 | 192 | 14 | Australia | 34 |
| National Yunlin University of Science and Technology | 13 | 190 | 5 | Netherlands | 22 |
| University of New Hampshire | 4 | 174 | 3 | New Zealand | 7 |
| National Taiwan University of Science and Technology | 8 | 166 | 7 | Iran | 18 |
| Delft University of Technology | 2 | 140 | 2 | Scotland | 8 |
| University of Wollongong | 2 | 132 | 1 | Canada | 16 |
| Autonomous University of Barcelona | 3 | 129 | 3 | Malaysia | 12 |
| Queensland University of Technology | 3 | 121 | 1 | Turkey | 7 |
| Wageningen University | 3 | 119 | 4 | South Korea | 12 |
| University of Glasgow | 3 | 107 | 1 | Belgium | 5 |
| University of Utrecht | 2 | 101 | 2 | Singapore | 5 |
| University of Amsterdam | 2 | 101 | 2 | Indonesia | 10 |
| Harbin Institute of Technology | 2 | 97 | 3 | Saudi Arabia | 13 |
| University of Illinois | 6 | 93 | 1 | Russia | 6 |
| University South Florida | 3 | 88 | 2 | Wales | 2 |
| Open University of Catalonia | 2 | 86 | 1 | Portugal | S2 |

### 4.3. Literature Clustering

The results without online-first publications (N = 457) were grouped by the "clustering" function in CitNetExplorer according to their citations. This function could analyze the relatedness of the publications and determine the cluster to which each publication belongs

according to the overall measurements of its citation relationships with other publications (Van Eck and Waltman 2017). As Figure 3 shows, four clusters were identified, while 188 publications did not belong to a group due to the minimum setting of group size (minimum size = 10). Group 1 (G1, N = 168) and Group 2 (G2, N = 140) took up 67.40% of all publications, reflecting the primary relationships in the literature search results. Group 3 (N = 16) and Group 4 (N = 12) were relatively small. Therefore, we primarily considered the two large clusters to further examine the contents of the literature.

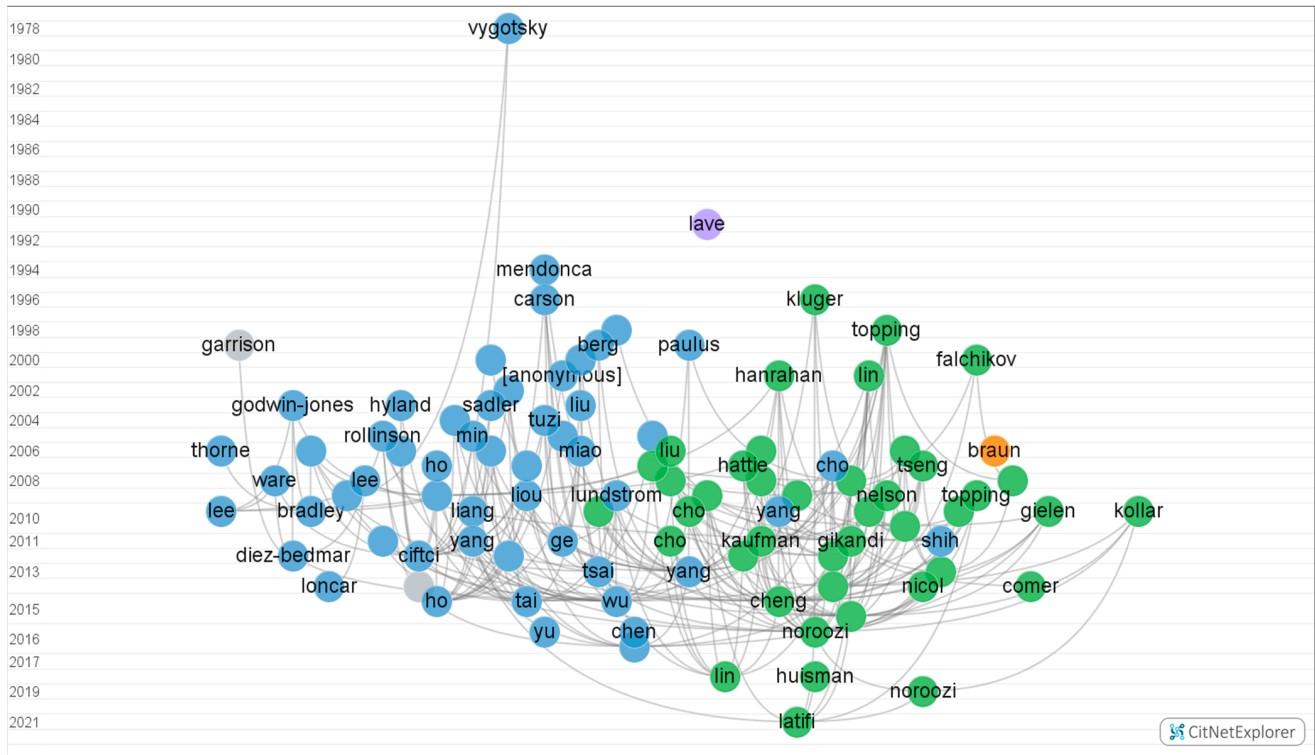

**Figure 3.** The citation network of publications.

RQ3 and 4 led us to the distinct research focuses between the two major groups (G1 and G2). We exported the clustering results of G1 and G2, identifying their interests in specific educational technologies, learning contents, and participants' educational levels by reading their titles, keywords, and abstracts. Studies in G1 mainly concentrated on the discussions on collaborative writing in online contexts (Tang et al. 2022; Sun and Zhang 2022; Zhang et al. 2022) with a small proportion of other language skills. Corresponding to the keyword clustering results in Table 3, topics related to writing skills have been frequently investigated in the search results. The studies on writing were extended to the virtual writing course (Payant and Zuniga 2022), academic writing (Zhang et al. 2022), writing revision according to automated and human feedback (Tian et al. 2022), and so forth. By contrast, G2 took a different perspective and attached the priority to learning outcomes of peer assessment and peer feedback in online language courses, dominantly examining the psychological and behavioral aspects. This cluster demonstrated the interdisciplinary research trends revealed in previous sections. Examples of interacting research areas could be found in the literature search results: Peer feedback could affect learners' motivations in gamified learning (Saidalvi and Samad 2019). Peer assessment was also related to reflexive practice (Shek et al. 2021).

4.3.1. The Theoretical Foundations of Peer Assessment

In order to examine the theoretical foundations and the related citation networks (RQ5), we analyzed the citation paths in G1 and G2. The most cited publication in all clusters was Vygotsky and Cole (1978), with citations = 40. Other highly cited publications included

Topping (1998) (in G1) with citations = 33, and Liu and Carless (2006) (in G2) with citations = 31. Vygotsky and Cole (1978), cited more than 70 times in the filtered results, served as the pioneer of the studies on peer assessment in language courses. In their *Mind in Society*, the theory of "Zone of Proximal Development" (ZPD) set the foundation of collaborative learning, suggesting learners make advancements as a result of their collaborative activity (Ali 2021; Vygotsky and Cole 1978). Similar to the idea of ZPD, Vygotsky's scaffolding theory indicated the importance of guidance suitable to learners' cognitive levels (Vygotsky and Cole 1978). Topping's article conducted an early review of peer assessment, providing a typology, benefits, theoretical foundations, validity, and reliability evaluations of peer assessment (Topping 1998). Previous studies also established that Topping's article was a pioneer in proposing peer assessment (Lin 2019). Strong evidence from a literature review and a large-scale questionnaire survey provided rationales for peer assessment that might enhance students' learning outcomes (Liu and Carless 2006).

Analyses of each group's citation network revealed the relationships after literature clustering. In G1 (studies in blue color in Figure 3), we selected the publications with the highest citation scores, including Vygotsky and Cole (1978), Lundstrom and Baker (2009), Yang et al. (2006), Min (2005), Min (2006), Cho and Schunn (2007), and Tuzi (2004). However, we found no citation path longer than 3. Recent studies, such as Van den Bos and Tan (2019), Saeed et al. (2018), Pham (2021), and Lv et al. (2021), directly cited the former group of publications, i.e., the longest citation path length = 1. Thus, citation network analysis was not suitable for this group. However, we found that the theoretical foundations represented by Vygotsky and Cole (1978) were still accepted in current studies.

In G2, the longest path was identified by selecting the most cited publication by Topping in 1998 and one of the recent studies, i.e., Latifi et al. (2021). We exhibited all intermediate publications between these two papers and identified 15 results between them. Multiple longest paths (length = 8 publications) between these two involved 11 publications and were marked in light yellow in Figure 4. The other four dark-green dots were also intermediate publications between the two selected ones, but not on the longest paths. Interestingly, we found that a group of common citations identified in these paths existed between multiple recent studies and Topping's article in 1998, as demonstrated in Table 6. Between Topping 1998 and each of the recent studies in the second column, all or most of the common citations in the third column appeared as intermediate publications.

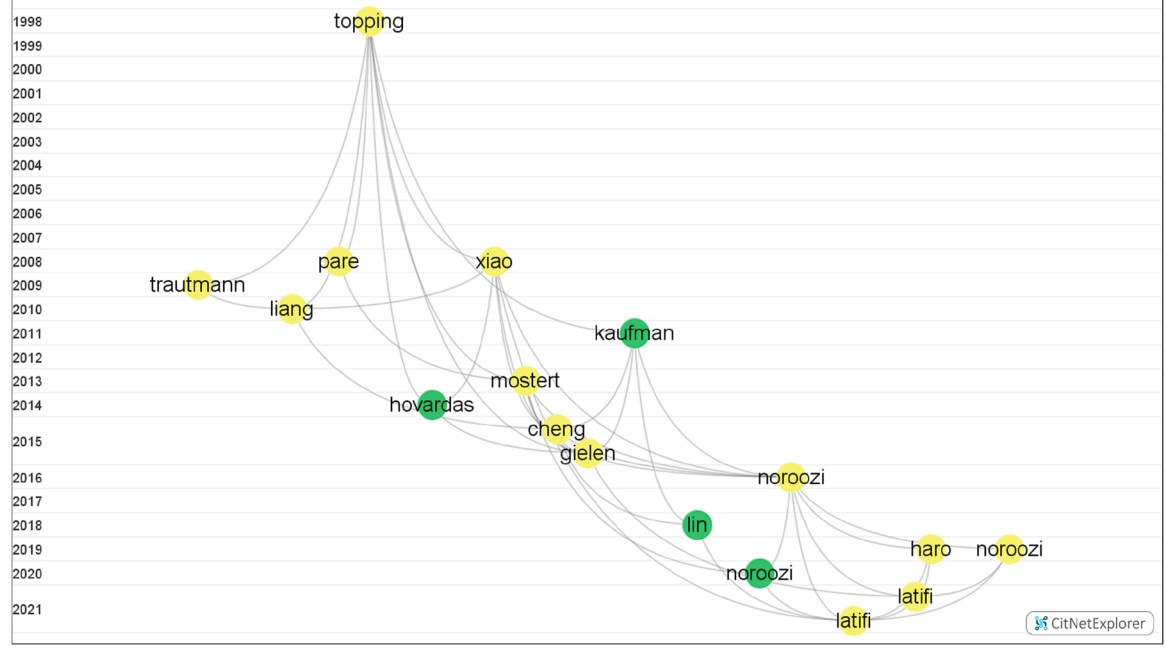

**Figure 4.** Multiple longest paths between Topping (1998) and Latifi et al. (2021).

**Table 6.** Common citations in multiple longest paths between Topping (1998) and eight recent studies published in 2021 and 2022.

| The Most Cited Publication | Recent Publications | Commonly Cited Publications on the Longest Paths |
|---|---|---|
| Topping (1998) | Haro et al. (2019)<br>Hoffman (2019)<br>Latifi et al. (2021)<br>Lin (2019)<br>Nicolini and Cole (2019)<br>Noroozi and Hatami (2019)<br>Noroozi et al. (2020)<br>Zheng et al. (2019) | Gielen and De Wever (2015); Liang and Tsai (2010); Mostert and Snowball (2013); Noroozi et al. (2016); Paré and Joordens (2008); Trautmann (2009); Xiao and Lucking (2008) |

### 4.3.2. Integration of Educational Technologies into Peer Assessment in Online Language Courses

We examined the most recent studies published in 2021 and 2022, including 31 studies in G1 and 37 in G2 identified by CitNetExplorer to answer RQ6. More specifically, we identified from these studies the educational technologies applied to peer assessment in online language courses. Similar to the identification of distinct focus above, studies in G1 indicated that teachers of online language courses developed multiple designs of collaborative learning, with peer assessment considered an easily performed and beneficial learning strategy (Lv et al. 2021). Academic writing, notetaking in lectures, and revision tasks could be enhanced with peer assessment (Zhang et al. 2021; Choi et al. 2021). Applied to synchronous computer-mediated online language courses, corrective feedback provided by peer learners could benefit grammatical knowledge learning for EFL students (Mardian and Nafissi 2022). Compared with automated writing evaluation technologies in self-regulated language learning, peer assessment and teacher feedback activated more cognitive and motivational strategies (Tian et al. 2022). Additionally, educational technologies in providing peer assessment in online language courses included the following: an online learning community for teaching writing (Tang et al. 2022), video peer feedback in a Chinese-English translation course (Ge 2022; Odo 2022), learning analytics (Chen et al. 2022), an academic English writing MOOC (Wright and Furneaux 2021; Fang et al. 2022), Google Docs (Ali 2021), and video annotation used for peer assessment (Shek et al. 2021).

## 5. Discussion

### 5.1. Publication Trends

A possible reason that publications about peer assessment in online language courses have seen a sharp increase around 2019 is the urgent need for distance education triggered by COVID-19 (Adedoyin and Soykan 2020), which requires active interdisciplinary research. In order to provide a more comprehensive understanding, researchers resort to perspectives of multiple research domains, such as psychology in education, cultural studies, and technological advancements. The integration of research domains and subjects also contributes to the formation of the interdisciplinary trend of peer assessment in online language education. Introducing theories and approaches in multiple research areas into language education research produces diversified focuses on peer assessment in online language courses. As found in the most used keywords, researchers referred to peer assessment as a collaborative learning strategy with slightly different terms. This demonstrates different pedagogical purposes in actual teaching practice. As Stovner and Klette (2022) suggested, assessment and evaluation are distinguished from feedback in that the former terms tend to be summative, while the latter is a formative approach to learning. Students can improve their further learning based on instructive and evaluative information in peer feedback, while assessment primarily emphasizes grading outcomes than constructive guidance. Such a distinction can explain why other terms are used in line with "assessment".

Higher education is a top-used keyword, while other educational levels are less investigated. This is probably because higher education contexts allow more flexible and diversified teaching designs or even teaching experiments. Higher education involves more diversified but specialized learning contents, where instructors actively seek effective methods to deliver learning contents to their students. Additionally, students' expertise is relatively limited before higher education levels, so they may not evaluate their peer performances from a comprehensive perspective of particular subjects. When faced with difficulties in academic contexts, higher education students can rely on more experience in exploring how to provide proper suggestions on improvements for their peer students. Similarly, based on academic expectations and experiences, higher-education students have stronger abilities to explore expertise in their subjects and think critically and independently than students under the higher education level. Peer assessment in higher education finds a balance between using its advantages and minimizing the risk caused by students' limited expertise. The challenges and the benefits exist simultaneously, while overemphasizing the risk may prevent peer assessment from wide application and further development in higher education (Ashenafi 2017). The solutions to some concerns are being provided by the updated and emerging technologies in recent years.

On the global scale, the shared research interest in language education encourages researchers to investigate peer assessment in online language courses, whether the studies be conducted in English- or non-English-speaking countries. English-speaking countries outnumber non-English-speaking countries in terms of publications probably because the literature search is based on English publications. It is reasonable to assume that researchers in various countries across the globe are devoted to localizing educational technologies in online teaching and learning their local languages. For English publications, studies on peer assessment in online language courses are not restricted to English-as-first-language contexts. They also include foreign language and second language teaching (see Ge 2022 as an example of the Chinese EFL learning context). Peer assessment in teaching languages other than English online can also be found (Tsunemoto et al. 2022). The enhanced learning effectiveness and benefits of peer assessment in online language education are supported by evidence from search results around the globe.

The clustering of the literature demonstrates a preference for studies on writing over other linguistic skills. This research trend may result from more convenient and practical attempts to implement peer assessment in online writing tasks than other language skills. First, comments on writing are clear and efficient, while it requires much more time to review peer performances in other forms, let alone the reviewers providing efficient comments and the receivers' understanding. For instance, performances in videos and spoken tasks are not as straightforward as written tasks regarding review and evaluation. Time is critical in implementing self-assessment and peer-assessment activities (Siow 2015). The time-consuming nature, with presumably limited accuracy, prevents students from accepting peer assessment and teachers from considering peer assessment activities. Therefore, the required time for peer assessment and the convenience of providing peer suggestions for improving language skills may be an important element in explaining the existing preference for investigating peer assessment in online writing courses. Second, broader and higher-level knowledge is required to identify problems in performances other than general writing tasks, including academic writing, thinking, learning strategies, oral presentation, and communication, and provide practical solutions to improvements (Zhang et al. 2021; Shek et al. 2021). In contrast, writing tasks for general purposes allow peer students to provide their understanding and suggestions even if the reviewers still need to develop their expertise in writing. The relatively common requirements can enhance the acceptance and usefulness of peer assessment activities.

## 5.2. Theoretical Advancements

According to Vygotsky and Cole (1978), students can easily learn what they can achieve under suitable guidance to their current levels of cognition and knowledge, which

explains the importance of peer support. Consequently, peer students with similar levels of knowledge and cognitive competence are the best sources of such guidance. This explains why peer assessment could be originated from his theory. Between his original theories and the current studies, a group of common citations identified in the above section delineate critical issues related to peer assessment: Engaging students in online peer assessment is an essential challenge and research topic; the roles of students who provide and receive peer assessment can be the key to the challenge (Gielen and De Wever 2015). Comparison between self-assessment, peer assessment, and expert assessment is a way to understand the unique advantages of peer assessment (Liang and Tsai 2010; Paré and Joordens 2008; Trautmann 2009). Peer assessment in online language courses shows advantages over traditional formats. In this sense, findings by comparison of online module-based and traditional paper-based peer feedback are significant in technology-assisted educational contexts (Mostert and Snowball 2013). Different peer assessment designs can also help to explore the enhanced approaches to successful peer assessment (Xiao and Lucking 2008). How peer feedback can succeed in terms of its quality is also a meaningful topic. High-quality, engaging, elaborated, and justified peer feedback leads to better writing outcomes (Noroozi et al. 2016).

The findings of the citation network analysis have led to the above critical research issues in the existing literature. Current studies are still interested in peer assessment in online language courses and widely cite findings about student engagement, unique advantages, enhanced designs, and critical factors for success. These may largely explain the citations used in the most recent studies about writing (Hoffman (2019); Lin (2019); more specifically, argumentative writing and learning in Latifi et al. (2021); Nicolini and Cole (2019); Noroozi and Hatami (2019); Noroozi et al. (2020)). The most recent studies are based on the identified common citations, since they have pointed out the above specific research directions and topics in this area.

*5.3. Integration of Educational Technologies and Interdisciplinary Trends*

The increasing use of educational technologies and the interdisciplinary research trends have been consistent with the influences of Web 2.0 on collaborative learning since around 2005 (Hegelheimer and Lee 2013). Following internet technologies, portable and function-specific technologies have emerged in the past two decades. This encourages peer assessment in online language courses to involve multiple educational technologies, with close investigations of recent studies published in 2021 and 2022 on the rapid updates and developments of educational technologies, as listed by Haleem et al. in 2022. The diversified educational technologies integrated into peer assessment are also consistent with previous reviews and meta-analyses regarding technology-assisted language learning (Burston and Giannakou 2022). As a learning and teaching strategy, peer assessment can now be realized by various technologies and can enhance multi-faceted language skill acquisition and linguistic knowledge in specific aspects.

Interdisciplinary trends in this area can explain why many most-used keywords belong to subjects outside language studies: studies combine peer assessment in online language courses and hot issues in psychology, including keyword items such as efficacy, engagement, perception, and motivation (see Table 3 and Figure 2). Similarly, one publication can be classified into multiple categories (see Table 1). Consequently, the percentages of the top publication categories amount to over 100 percent due to multiple counts for the same publication in different categories. Studies on peer assessment in online language courses now involve multiple subjects and research areas. By comparing the predominant technologies and those applied to peer assessment in online language courses, factors such as instructors' technology literacy, support from policies and regulations, and the effectiveness of different technologies can impact the actual application of peer assessment in online language courses. The acceptance of educational technologies has been another popular research topic, especially with the wide adoption of structural equation modeling (Zhang and Yu 2022). An increasing number of factors are being included in explaining

this mechanism whereby particular educational technologies can be accepted by learners and instructors and adopted in teaching practice.

Based on recent investigations on educational technologies and interdisciplinary research trends, the integration of educational technologies into peer assessment in online language courses should at least consider the following aspects. First, functional characteristics of technologies (such as interactivity, the timing and pacing of the learning process, and simulated or immersive experience) would determine whether students will significantly benefit from technology applications to particular learning contents. When the characteristics of the learning contents can be enhanced by the advantages of the educational technologies, the matched technologies and contents will more likely promote students' learning outcomes. For example, demonstration through video feedback may be more useful than other technologies for skill learning where simulation and visual aids matter (Shek et al. 2021).

Second, diversifying designs and updating features of the existing educational technologies may provide solutions to previous limitations, which requires the instructors to stay on pace with the advancements of educational technology developments. Some educational technologies still need validation through practical experience of peer assessment applications in online contexts. Through comparing the most recent studies and a broader framework of educational technologies proposed by Hallem et al. in 2022, the enhanced effectiveness of peer assessment has been well established for technologies in online language education, such as mobile applications (Chang and Lin 2020) and web-based learning communities (Lai et al. 2019). However, with the unbalanced attention attached to different technologies, some may need greater attention in order to reveal their usefulness in peer assessment in online language courses. As an emerging technology, artificial intelligence chatbots have recently been introduced into language education, though their effectiveness remains controversial (Huang et al. 2022). Learning analytics has received keen interest and helped understand students' perceptions of peer assessment (Misiejuk et al. 2021), but only limited literature can be found. For less-investigated technologies, instructors may need to consider the features and functions of the technologies, i.e., compare how characteristics of particular educational technologies fit the teaching contents of the targeted language skills.

Third, according to studies on the influencing factors in subjects outside language education and linguistics, instructors should consider psychological, social, cultural, gender-related, and other factors. Educational technologies can demonstrate varying effects for different participants (Yu et al. 2022a, 2022b). This principle, that multi-faceted factors should be investigated and considered, is consistent with some existing studies that aimed to provide an understanding of influencing factors on online learning. For example, Yu et al. (2022a) identified influencing psychological factors on MOOCs, including learning engagement, students' motivation, perceptions, and satisfaction. Wu and Yu explored achievement emotions in online learning (Wu and Yu 2022). Learning strategy can be diversified in mobile English learning (Yu et al. 2022b), and it may also be a critical factor in peer assessment in online language earning. Such interdisciplinary studies evaluating technology-enhanced education provide an understanding of the influencing factors in educational technologies. According to these studies, solutions may be found to diversify the application of educational technologies to online language courses and enhance the effectiveness of online peer assessment.

## 6. Conclusions

### 6.1. Major Findings

This bibliometric analysis used visualization and citation network analysis to investigate peer assessment in online language courses. We found that this topic has witnessed a sharp increase in publications since 2019 and remained popular. The publications also demonstrated international and interdisciplinary trends by introducing concepts and approaches in multiple research areas, as well as emerging educational technologies. Peer assessment was not a new learning and teaching strategy that occurred way earlier than on-

line learning. Despite theoretical foundations from more than a century ago, recent studies developed and identified specific research topics and important findings, including critical factors for improving students' engagement and feedback quality, unique advantages in online peer assessment, and designs to enhance peer assessment quality. Current studies preferred to investigate writing skills with online peer assessment, but other integrative and complex linguistic skills were also studied in terms of the effects of peer assessment. Another robust research trend was that current studies had strongly welcomed studies involving emerging educational technologies. International and interdisciplinary research trends and the increasingly popular integration of educational technologies may diversify peer assessment in online language courses and enhance the effectiveness of online peer assessment activities.

### 6.2. Limitations

Some limitations have to be acknowledged regarding the methods of visualization, citation network, and bibliometric analyses. Compared with systematic reviews and meta-analyses, bibliometric analyses could not dig into details of the literature, even if it might include a much larger size of literature at a time. However, in order to address such primary limitations, this bibliometric analysis study not only utilizes the vivid demonstration of authors, keyword items, organizations, countries, and studies, but also examines some frequently cited and most recent studies on this topic to reveal the frontier research topics and the research trend. This study is subjected to some limitations specific to this study. First, this study conducted a literature search in English publications due to the limitation of our linguistic knowledge. Studies published in other languages were not included in this study. Second, the exploration of theoretical foundations relied on the literature search results and did not fully take a chronological perspective in tracking the origin of the related theories.

### 6.3. Implications for Future Research

Theoretically, the critical research directions identified in citation network analysis will continue encouraging studies to contribute to those research issues. The overall goal will still be to improve the understanding of peer assessment in online language courses, ultimately facilitating language teaching in the era of e-learning. More practically, future studies may extend the studies on peer assessment in online language courses in multiple directions. Localization of peer assessment encourages researchers across the globe to explore and establish the effectiveness of peer assessment in their native languages other than English. Educational technologies are emerging and updating quickly, encouraging researchers to integrate some less-studied technologies into peer assessment in online language courses. Peer assessment, in proper forms, may also be investigated at other educational levels, extending teaching practice and the experience of implementing peer-assessment approaches. Future studies that aim to review studies on this topic may adopt other approaches, such as meta-analysis and systematic review. These two methods may lead future researchers to more specific analyses in this area. Studies on peer assessment under online learning contexts will also shed light on the further application of educational technologies and enhance teaching and learning outcomes.

**Author Contributions:** Conceptualization, Y.L. and Z.Y.; methodology, Y.L.; software, Y.L.; validation, Y.L. and Z.Y.; writing—original draft preparation, Y.L.; writing—review and editing, Y.L. and Z.Y.; visualization, Y.L.; supervision, Z.Y.; project administration, Z.Y.; funding acquisition, Z.Y. All authors have read and agreed to the published version of the manuscript.

**Funding:** This research was funded by the 2019 MOOC of Beijing Language and Culture University (MOOC201902) (Important) "Introduction to Linguistics"; the "Introduction to Linguistics" of online and offline mixed courses in Beijing Language and Culture University in 2020; the special fund of the Beijing co-construction project—research and reform of the "Undergraduate Teaching Reform and Innovation Project" of Beijing higher education in 2020—an innovative "multilingual+" excellent

talent training system (202010032003); and the research project of the graduate students of Beijing Language and Culture University, "Xi Jinping: The Governance of China" (SJTS202108).

**Institutional Review Board Statement:** Not applicable.

**Informed Consent Statement:** Not applicable.

**Data Availability Statement:** Not applicable.

**Acknowledgments:** The authors would like to extend sincere gratitude to their funders, journal editors, anonymous reviewers, and all other participants who offered help to review and publish this paper. The first author would like to deliver special thanks to his supervisors in the master's program.

**Conflicts of Interest:** The authors declare no conflict of interest.

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
