# Peer review of "A Bibliometric Analysis of Peer Assessment in Online Language Courses"

_languages, doi:10.3390/languages8010047_

Round 1

Reviewer 1 Report

Summary

The manuscript is a bibliometric analysis of recent publications on peer assessment in online language courses. The study identifies current publication trends and citation networks on the basis of visualisation and citation network analysis. More specifically, the authors show that peer assessment in online courses has received increasing attention since the COVID-19 pandemic, with research foci becoming more diversified and interdisciplinary. As such, the paper makes a useful contribution to the field, potentially helping others to identify avenues for future research.

General comments

The manuscript has several merits that are worth highlighting:

·         The study integrates peer feedback with educational technologies, two key topics in current language teaching.

·         It identifies core research topics, authors, and other aspects of the pertinent literature, as well as their relationships, based on a considerable number of publications.

·         It provides a useful basis for better understanding the overall intellectual landscape relating to the topic.

·         The paper has a clear overall structure.

Areas for improvement:

·         The Literature Review section is not a literature review in the conventional sense. Instead of discussing relevant sources in conversation with each other, it provides a rough outline of some observed trends in the literature, interspersed with research questions and elements of a ‘roadmap’. The authors may want to reconsider the section heading so as not to confound readers’ expectations.

·         The Literature Review section remains somewhat vague and superficial at times, e.g.

o   “… empirical results on this topic have been significantly enriched by various perspectives, theories, technologies, and approaches” (lines 68-69). What perspectives, theories, technologies, and approaches are the authors referring to?

o   Different authors, organizations, and journals showed rising interest in peer assessment in online language courses” (lines 71-72). Which authors, organisations, and journals do the authors mean?

o   “The development of applied linguistics by involving theories and concepts from various subjects and research areas made interdisciplinary approaches increasingly crucial in current studies” (lines 91-93). Which theories, concepts, subjects, and research areas are the authors referring to?

Further support or illustration would be useful here.

·         The second paragraph in the Literature Review section lacks development. I would recommend working on the coherence and cohesion in this paragraph.

·         The heading of Section 3 is inappropriate (Results vs. Methods).

·         Readers who are not fully familiar with WOS, VOSviewer and CitNetExplorer might benefit from some more general information about these tools in the methods section.

·         The Results section requires greater clarity in several respects:

o   The large amounts of numerical data in the Results section are difficult to digest. To enhance the readability and comparability of this section, I would strongly recommend presenting the results in tabular form whenever possible and appropriate, not as part of the running text.

o   Section 4.2 moves from keyword clusters to authors and back to keywords. Would it make sense to sequence the information differently? What can we make of the keyword clusters? Three out of 17 clusters are mentioned. What about the others?

o   The reader would benefit from more effective integration of the figures into the running text and a more detailed description of what the figures actually show, for example, how should Figure 2 be interpreted and what are the distinct benefits of this type of visualisation?

o   Readers who are not familiar with the clustering function in CitNetExplorer would benefit from some more general information on that function. And how exactly was the research focus identified?

o   RQ3 and RQ4 are relatively poorly developed in the Results section. The authors may want to elaborate on them.

o   The methodology used to answer RQ6 is not quite clear to me.

o   The visual quality of the figures needs improvement (esp. resolution and font size).

·         Some parts of the Discussion section lack clarity. For example, the second paragraph on higher education requires interpretation on the reader’s part (esp. lines 399-406). The first paragraph in Section 5.2 lacks coherence and development (lines 438-456). The four aspects mentioned in section 5.3 (lines 491-497) need further explanation.

·         The manuscript mentions two important limitations of the study. However, it does not discuss the limitations of bibliometric analysis more generally. The authors may want to address some disadvantages of this type of analysis, possibly in comparison with more traditional approaches such as systematic literature reviews.

·         There are several language problems, which require re-reading and interpretation on the reader’s part. The manuscript will benefit from careful proofreading.

Specific comments

·         Lines 53-61: The authors may want to reconsider some specific uses of modality (“We would first review …”, “The findings of this study would provide …”) and tenses (“This bibliometric study focused …”).

·         83-85: I would recommend rephrasing for the sake of clarity.

·         88: English for Specific Purposes is usually not considered to be a “skill”.

·         89-91: Meaning unclear

·         93-95: I would recommend rephrasing for the sake of clarity.

·         114-116: I would recommend rephrasing for the sake of clarity.

·         206: Is “rapid increase” the most precise expression here (considering that Figure 1 actually shows a decrease in 2020)?

·         The authors may want to be consistent in the number of decimal places they report throughout the text.

·         240: citation scores vs citations?

·         334: acronym not introduced

·         358, 360: boldface?

·         382-384: I would recommend rephrasing for the sake of clarity.

·         419-421: I would recommend rephrasing for the sake of clarity.

·         425ff: The claim that peer assessment in online writing tasks may be more convenient and practical needs clearer explanation/support.

·         434-436: Meaning unclear

·         481: needs explaining

·         498ff: This paragraph remains quite vague (“some technologies …”, “such technologies …”, “different technologies …”). The authors may want to be more specific here.

·         512: acronym not introduced

Author Response

Dear Reviewer, 

We have considered all your comments and revised our manuscript. Please see the attachment for point-to-point response.

Kind regards.

Reviewer 2 Report

This manuscript presents a very strong paper with well-defined research with conclusive results. All the sections of the paper are written with rigor and it follows academic guidelines. All references are correct and no editions in terms of language are required. However, I wonder why in the literature review authors include a rationale for RQs 2, 4 and 5 and there is no information about questions 1, 3 and 6. For consistency, I suggest that authors include information regarding these questions in the literature review section. Alternatively, they can organize the literature review and then include a Methodology section where they list the Research Questions. 

Author Response

(The authors gave the same response as above.)

Round 2

Reviewer 1 Report

Thank you for considering my feedback. I think that the manuscript has improved greatly in terms of clarity, coherence, and readability. There are still some minor issues (wrong heading for Section 3, wrong caption for Table 5, numbering of tables, use of tenses, etc.) which require careful editing.

Author Response

Dear Reviewer,

We have corrected the following mistakes (We referred to the submitted revised version for the following line numbers):

  1. Heading of Section 3: "Results" has been changed to "Methods".
  2. Caption of Table 5: The duplicated part has been changed to "top 20 organizations and countries".
  3. Numbering of Tables checked: "Table 1" (Line 345), demonstrating the commonly cited group of studies, has been changed to "Table 6". An in-text reference to this table (Line 340) has been changed to the consistent number.
  4. Logical structure of Section 5.3 improved.
  5. Full expression of "MOOC": Moved from Line 532 to 131 where it first appeared in the article.
  6. Redundant blank line (347) has been deleted.

We have also examined citation formats (especially several minor editings in the reference page), grammar and spelling for this round. We hope the revision will solve your concern.